# The use of heuristics in genetic testing decision-making: A qualitative interview study

**Bettina Maria Zimmermann**[1,2]*, **David Martin Shaw**[1,3], **Bernice Elger**[1,4], **Insa Koné**[1]

**1** Institute for Biomedical Ethics, University of Basel, Basel, Switzerland, **2** Institute of History and Ethics in Medicine, School of Medicine, Technical University of Munich, Munich, Germany, **3** Care and Public Health Research Institute, Maastricht University, Maastricht, The Netherlands, **4** Center for Legal Medicine, University of Geneva, Geneva, Switzerland

* bettina.zimmermann@unibas.ch

## Abstract

### Background

Decision-making concerning predictive genetic testing for hereditary cancer syndromes is inherently complex. This study aims to investigate what kind of complexities adults undergoing genetic counseling in Switzerland experience, how they deal with them, and what heuristics they use during the decision-making process.

### Methods

Semi-structured qualitative interviews with eighteen Swiss adults seeking genetic counseling for hereditary cancer syndrome genetic testing and two counseling physicians were conducted and analyzed using a grounded theory approach.

### Results

Counselees stated that once they were aware of their eligibility for genetic testing they perceived an inevitable necessity to make a decision in a context of uncertainties. Some counselees perceived this decision as simple, others as very complex. High emotional involvement increased perceived complexity. We observed six heuristics that counselees used to facilitate their decision: Anticipating the test result; Focusing on consequences; Dealing with information; Interpreting disease risk; Using external guidance; and (Re-)Considering the general uncertainty of life.

### Limitations

Our findings are limited to the context of predictive genetic testing for hereditary cancer syndromes. This qualitative study does not allow extrapolation of the relative frequency of which heuristics occur.

### Conclusions

The use of heuristics is an inherent part of decision-making, particularly in the complex context of genetic testing for inherited cancer predisposition. However, some heuristics

**Data Availability Statement:** This study is subject to the Swiss Human Research Act (HRA), which requires approval by a governmental ethics commission (art. 45). This study was approved jointly by the cantonal ethics committees Bern and

Northwest and Central Switzerland (approval number 2017-00316). The function of this ethics commission according to HRA is to approve that the study design and execution is in line with the legal, ethical and scientific standards of the HRA. According to HRA art. 59 §4b, personal data can only be shared is there is specific written consent is given by participants. For this study, participants did not give such a consent. The consent form states that encrypted transcripts of interviews are only shared within the research team conducting the study and, if necessary for quality assessment, with the ethics committee. Therefore, the full interview transcripts cannot be shared outside the Institute for Biomedical Ethics, University of Basel. Moreover, the data sharing policy for this study as approved by the ethics committees states: "Data will be accessible to third parties through the publications only in an anonymized form." Since full transcripts cannot be fully anonymized due to the highly individual context, including personal family history, rare genetic mutations as well as the recruitment on two sites only, full transcripts cannot be shared with anybody outside the research team. Any requests concerning data access can be directed to a.loschnigg@unibas.ch.

**Funding:** BMZ: Freie Akademische Gesellschaft Basel (personal career grant for completion of PhD thesis, no grant number assigned). https://fag-basel.ch The funder did not play any role in the study design, data collection and analysis, decision to publish, or preparation of the manuscript.

**Competing interests:** The authors have declared that no competing interests exist.

increase the risk of misinterpretation or exaggerated external influences. This may negatively impact informed decision-making. Thus, this study illustrates the importance of genetic counselors and medical professionals being aware of these heuristics and the individual manner in which they might be applied in the context of genetic testing decision-making. Findings may offer practical support to achieve this, as they inductively focus on the counselees' perspective.

## Introduction

Individuals suspected to carry genetic variants that lead to hereditary forms of cancer face the decision of whether to take a genetic test [1]. Performed in unaffected individuals, such tests provide a predictive assessment of the risk of developing cancer in the future and can guide preventive measures, such as regular screening, preventive medication, or surgeries. However, such predictive genetic tests also hold important psychosocial implications, can lead to anxiety, and might also indirectly provide risk information for relatives [2]. Moreover, the interpretation of predictive genetic test results is challenging, as cancer risk assessments depend on numerous factors such as the exact genetic variant, family history of cancer, and clinical evidence available for the particular variant [3]. Even a genetic test that does not detect any particular pathogenic genetic variant does not imply that the tested individual is free of genetic risk factors for developing cancer [4, 5]. Finally, the nature of the test's consequences also depends on many individual factors and involve more medical decisions for the affected individual after the genetic test. For example, women carrying pathogenic variants in the *BRCA1* or *BRCA2* genes can opt for regular breast cancer screenings or preventive mastectomies to lower cancer risk [6, 7]. The decision-making process in genetic testing for cancer predisposition and other forms of predictive genetic testing is, therefore, considerably complex.

In anticipation of this complex decision-making process, genetic counseling usually accompanies predictive genetic testing in the clinical context. In many countries (including Switzerland) genetic counseling is mandatory before and after predictive genetic testing [8]. Genetic counseling is expected to be performed in a non-directive manner, which in Switzerland is legally determined in the Human Genetic Testing Act (art 14/1). Thus, the counseled individual has a free choice in deciding whether to accept or decline predictive genetic testing.

From an ethical perspective, such a decision should be made intentionally, with an understanding of all relevant available information and in absence of significant external controlling influences [9]. The concept of nondirectiveness in genetic counseling attempts to avoid the imposition of external controlling influences via the healthcare professional, but this principle has been debated since the counseling process as social interaction is inherently influential [10–12]. Still, genetic counselors have an ethical duty to be aware of their potential influence and avoid coercive or manipulative situations. To meet the requirements of intentionality and understanding, the cognitive process of processing factual information is an inherent part of informed decision-making. However, too much information in complex decision-making processes might lead to confusion and information overload [13, 14]. Heuristics, defined as "mental shortcuts or 'rules of thumb' that decision-makers consciously or unconsciously employ to make judgments of uncertainty" [15], are commonly applied in such complex decision-making processes to avoid overload and facilitate decision-making [16–18]. Theoretical heuristics models distinguish between affect heuristic (the emotional distinction between "good" and "bad"), the representative heuristic (comparison with stereotypes and other schemata), the

availability heuristic (reliance on existing knowledge and experience), and anchoring heuristic (the orientation on existing numbers of comparison) [15].

Predictive genetic testing for hereditary breast or ovarian cancer syndromes and hereditary colorectal cancer syndromes (such as Lynch syndrome or hereditary polyposis syndromes) is widely established in the clinical setting. A variety of risk-increasing genetic variants have been identified in this context [19, 20]. They are all not fully penetrant, meaning that a predictive diagnosis of these variants does not necessarily lead to disease outbreaks. In breast cancer, high penetrance genetic variants mean that more than 50% of carriers develop the disease; below 20%, they are considered low penetrance genes; and between 20–50% moderately penetrant [21]. Preventive surgeries or treatments are usually not recommended for carriers of low or moderately penetrant genes [22]. In Switzerland, genetic testing related to hereditary cancer syndromes currently involves targeted or panel testing [23], or, more rarely, exome-wide analyses [24].

In this paper, we aim to analyze how people undergoing genetic counseling in Switzerland deal with the complexity involved in deciding for or against genetic testing for cancer predisposition. We interviewed at-risk adults who attended genetic counseling and decided upon genetic testing for heritable cancer syndromes (hereafter: counselees). We asked them how they experienced the decision-making process and found that their decision was heavily influenced by their life philosophy: while some like to leave life to fate, others prefer control, and genetic testing is perceived as taking control [25]. We also found many instances where counselees talked about how they dealt with the complexities inherent in such a decision-making process. Therefore, we here aim to present a more in-depth analysis of this aspect and investigate the following research questions: What complexities do counselees experience during the predictive genetic testing decision-making process? How do they deal with those and what is the impact of heuristics on the decision-making process? We then seek to extrapolate implications for informed decision-making and how the use of heuristics could be fruitfully addressed in the counseling process.

Most studies investigating heuristics in medical decision-making use hypothetical scenarios and a quantitative approach [26]. Qualitative inquiries that implicitly or explicitly cover the use of heuristics in the genetic testing decision-making progress focus on the UK [27], Canada [28, 29], or the United States [30]. Most qualitative studies distinctively focused on risk perception [31, 32] or psychosocial implications [33, 34]. In contrast to previous inquiries on the role of heuristics in the genetic testing decision-making process, our study takes an inductive approach to investigate the influence of heuristics and focuses on the implications for informed decision-making in the context of genetic testing for inherited cancer risk.

## Methods

This work is a secondary analysis from a grounded theory study about the genetic testing decision-making process in the Swiss context, where we found that reducing complexity was an integral part of this process [25]. We applied the COREQ guidelines for reporting qualitative studies [35] (S1 File). We performed semi-structured interviews with at-risk adults who went to genetic counseling regarding genetic testing for hereditary cancer predisposition (thereafter: counselees). To verify early findings and for data triangulation purposes, we additionally interviewed two medical doctors who provided genetic counseling to some of the counselees (thereafter: experts). All participants gave consent before the interview: counselees signed an informed consent form, and experts gave their informed consent verbally, in line with research ethics committee recommendations and the Swiss Human Research Act. The study was approved by the ethics committees of Northwest and Central Switzerland and Bern (No 2017–00316).

## Recruitment and data collection

Healthcare professionals providing genetic counseling at the University Hospitals Basel and Bern recruited eligible counselees by handing over the study information leaflet. The exact recruitment procedure varied according to the healthcare professional's preference: for four individuals, the healthcare professional provided the interviewer with the counselees' email addresses or phone numbers to contact them directly. In all other cases, individuals interested in participation contacted the interviewer proactively. One potential participant dropped out after initial contact for family reasons. Counselees between the age of 18–70 that were not considered vulnerable due to mental state or pregnancy by the ethics committees that approved the study were eligible. We included both cancer patients and individuals at risk of a hereditary cancer predisposition who attended genetic counseling and had a medical indication for genetic testing according to medical guidelines, such as cancer diagnosis at an early age or significant family history [36, 37]. During the recruitment process, we found that the views of males, healthy individuals, and those deciding against genetic testing were underrepresented yet potentially important for theoretical saturation. We therefore purposefully recruited these participants in a second part of the study. We applied a pragmatic approach to data saturation and critically examined the saturation of themes during the data analysis process [38].

The lead author conducted all interviews between August 2017 and February 2019. Depending on counselee preference, they took place at the University of Basel, Bern University Hospital, or at counselees' homes. Counselees and the interviewer did not have an established relationship before this study but talked over the phone to discuss the study, its implications, and any remaining questions from participants. They were alone when conducting the interview.

The semi-structured interview guide included questions about the counselees' experiences during the genetic testing decision-making process. The interview guide was pilot tested with two healthy volunteers (not included in the study) and adjusted after six and again after eleven counselee interviews based on preliminary findings (see S2 File for the counselee interview guide). We also presented three hypothetical scenarios of genetic testing decision-making and asked them how they would decide in these scenarios, and why. Scenarios were developed to address and enrich counselees' reflections on risk perception and medical actionability (scenario 1), the relative importance of genetic testing in counselees' lives (scenario 2), and environmental influence on disease outbreak (scenario 3). Field notes were taken before and after each interview.

Expert interviews were conducted after an initial analysis of all counselee interviews and included questions about issues that emerged during analysis, including questions on uncertainty and risk perception (see S3 File). The purpose of expert interviews was to add the experts' perspective to the analysis and to compare preliminary findings with expert knowledge. Because no new aspects came up, data collection of expert interviews was completed after two interviews.

All interviews were recorded and transcribed *ad verbatim*. Interviews in Swiss German dialect were translated into standard German upon transcription and all transcripts were pseudonymized. Transcripts were not returned to participants. For publication purposes, illustrative quotes were translated into English by a native German speaker (IK) and double-checked for accuracy by two co-authors (BMZ, DMS).

## Data analysis

We followed a grounded theory approach for data analysis [39, 40] and started the inductive analysis during data collection. MaxQDA 2018 (VERBI GmbH) was used for analysis.

Interviews were analyzed by open coding and discussed and reflected on alternative interpretations in memos and group discussions throughout the analysis. By connecting codes and emerging themes with each other, BMZ and IK collaboratively started building concepts and categories. At a later stage, we abductively built hypotheses and tested them on interview data, using the grounded theory coding paradigm [39]. For the here-presented secondary analysis, we took a thematic approach to develop a more in-depth analysis on one of the aspects identified in that interpretive process, aiming to identify patterns and issues related to the use of heuristics. We then critically reexamined the interviews for alternative explanations and additional aspects of the phenomenon. With the expert interviews, we also sought to identify potential additional aspects and alternative explanations. We conducted these interviews after an initial analysis of all counselee interviews and included them in the analysis to examine whether experts supported or contradicted our general understanding of interview data. While all participants commented on the original grounded theory study [25], no additional feedback was sought for this secondary analysis.

## Findings

A total of 18 counselees and two experts were interviewed for this study. Their demographic characteristics are presented in Table 1. Interview duration ranged from 27 to 101 minutes (mean duration 59 minutes). The findings first develop how counselees experienced the complexity of genetic testing decision-making and then present six heuristics that counselees commonly described when reflecting on their genetic testing decision-making process. Expert interviews were used to confirm and contrast counselees' views. Heuristics were perceived in various ways; we report these variations in the subsections of each heuristic. A detailed coding framework is available in S4 File.

**Table 1. Demographics of counselees (n = 18).**

| | |
|---|---|
| **Gender** | |
| Female | 14 (78%) |
| Male | 4 (22%) |
| **Age** | |
| 20–29 | 1 (6%) |
| 30–39 | 5 (28%) |
| 40–49 | 3 (17%) |
| 50–59 | 5 (28%) |
| 60+ | 4 (22%) |
| **Test result** | |
| Variant-positive | 7 (39%) |
| Variant-negative | 8 (44%) |
| Decided against genetic testing | 3 (17%) |
| **Cancer syndrome that was tested for** | |
| Lynch syndrome | 2 (11%) |
| Hereditary breast/ovarian cancer | 16 (89%) |
| **Other characteristics** | |
| At least one blood-related child | 12 (67%) |
| Cancer diagnosis at point of counseling | 6 (33%) |
| Previously detected pathogenic variant in the family | 5 (28%) |

## Experiencing complexity

The counselees' narratives illustrated their experience of complexity and the degree to which they perceived genetic testing decision-making as complex. Counselees stated that once they were aware of their eligibility for genetic testing they had to make a decision either for or against the test. They perceived an inevitable necessity to make a decision in a context of uncertainty. Some questioned whether and when genetic testing was a perceived (medical, psychological, or life-planning related) benefit compared to leaving one's destiny to fate.

> *The question for me is, from an ethical perspective, how much do we need to know about ourselves? Why not leave this to fate? There are two sides to this. (counselee 13)*

Perceptions of complexity varied between counselees. While some counselees underlined the difficulty in interpreting a positive or negative test result and in understanding what impact it would have on their lives, others described their decision to be tested as a logical consequence based on the availability and their eligibility of being tested.

> *We've had family reunions from time to time with my cousins and for some, it's still an issue, even though we've known for a couple of years [that a pathogenic genetic variant runs in the family]: 'Maybe we will get tested anyway'. Others say strictly: 'No, I don't want to know'. How do you judge that when you have a clear stance and the person in front of you has a different one? It's just been difficult. For me, it was clear [that I wanted to do the test], and for others, it might be just as clear that they do not want to know. (counselee 14)*

> *So, for them [the counselee's daughters] it was clear that if I had it [the pathogenic genetic variant], they would take the test right away. [...] With us it's just like that: If something is up, we will do whatever we can to make up for it. (counselee 17)*

Emotional involvement seemed to affect the perception of complexity. For example, one counselee explained how the term "mutation" in one of the *BRCA* genes triggered for her the need for breast removal, which raised negative emotions. By contrast, the term "variant" (used to describe low prevalence genes in genetic counseling, according to this counselee) did not incur such strong emotions, and the decision to analyze such low prevalence genes was perceived as much simpler by this counselee. Conversely, one expert reflected on the difficulty of explaining these variants to counselees in the context of panel testing, stating that those lower risks or variants of unknown significance added complexity that was difficult to communicate. This expert feared that counselees would instead overestimate the risks of these variants.

Another emotionally challenging aspect for some counselees, confirmed by experts, was their family history of cancer. While some reported that they were talking openly about both cancer and its hereditary component, other counselees were reluctant to ask questions regarding family members who died from cancer, because they were afraid of the negative emotions this would generate or because they found it strange to contact distant family members for rather personal questions. For some counselees, this was an important barrier to informed decision-making.

> *Yes, I had a hard time [asking my family for information]. I didn't know how to ask my aunt, whether to call or go by or write an SMS, whether I should say why I was asking or just say that I was [generally] interested. [...] You don't know exactly what you might trigger in that person when coming up with these questions [...]. (counselee 11)*

## Observed heuristics and their implications on the decision-making process

We observed six heuristics counselees adopted to deal with the aforementioned complexities: (1) Anticipating the test result, (2) Focusing on consequences, (3) Dealing with information, (4) Interpreting disease risk, (5) Using external guidance, and (6) (Re-)Considering the general uncertainty of life. These heuristics have different implications on the decision-making process (see Table 2 for an overview of the observed heuristics and their impact on counselees' decisions).

**Anticipating the test result.** Counselees built expectations concerning the outcome of their genetic test. Those who already knew that a close family member was a carrier of a pathogenic genetic variant referred to their 50:50 chance of having inherited the pathogenic variant. One counselee stated that knowing his mother was a carrier made him cope better with the test result (which turned out to be variant-positive) and also made him more rational during the decision-making process.

For those who were the first in their family to get a test this heuristic was particularly important as they expressed certain expectations:

*Somehow, I almost expected that I could carry it [the pathogenic genetic variant], too. I didn't think, I'm going to do this test and I'll be negative anyway, but rather thought: well, maybe I'm positive. (counselee 02).*

*And when I look at my pedigree I know that the risk [of cancer] is certainly increased. (counselee 11).*

This preconception influenced and guided the genetic testing decision-making process. Perceiving the risk of carrying a pathogenic genetic variant as low made it easier to decide against genetic testing. Some counselees based this expectation on the pedigree analysis which is part of genetic counseling, while others had a rather intuitive feeling about it.

**Focusing on consequences.** As confirmed by both experts, genetic counselors are supposed to inform counselees very cautiously about the consequences of genetic testing. Accordingly, many counselees had a clear idea about the next steps to take if the genetic test was positive, e.g. preventive surgery or regular preventive check-ups. It was important to them to think through these consequences before taking the test. To be aware of a positive test result and not take the suggested action seemed a higher psychological burden in terms of fear and uncertainty to some counselees than not taking a genetic test at all.

**Table 2. Overview of the observed heuristics and their impact on the decision-making process.**

| Observed heuristics | Influence on the decision |
|---|---|
| Anticipating the test result | Perceived low risk for positive test -> test no |
| | Perceived high risk for a positive test -> test yes |
| Focusing on consequences | Positive test -> preventive action is taken (e.g. mastectomy, regular preventive examinations) |
| | Negative test -> health care can go back to "normal" |
| Dealing with information | Postponing the decision |
| Interpreting disease risk | Polarized 'all-or-nothing' risk perception might lead to anxiety or carelessness |
| Using external guidance | Using the example of others when deciding for or against genetic testing |
| (Re-)Considering the general uncertainty of life | Having the impression that own choices have only a limited impact on health status/life span |

*So that's why I am saying, with this test, you have to know very clearly beforehand [. . .] would you agree to have both breasts amputated, and would you have your ovaries removed at the age of 40? (counselee 09)*

Thinking through eventual results and consequences, counselees seemed to develop a personal decision tree, which guided them along the testing process. For example, one counselee stated:

*I really expected to have it [the pathogenic genetic variant]. Because of the whole family history [. . .]. And I had already planned step by step how to continue [. . .]. I would have done the surgery to remove my breasts, just for prevention, that would have been my next step. (counselee 03)*

Consequently, counselees focusing on consequences were reluctant to take a genetic test in the absence of medically actionable consequences in case of a variant-positive test result.

**Dealing with information.** Those counselees who felt overwhelmed by the complexity of the decision tended to escape the situation by avoiding information on the subject. Young individuals, for example, postponed their decision regarding whether to take a test until consequences such as screening programs or preventive surgeries were recommended due to age.

*And I was much younger back then and inexperienced, and I found it difficult. And then with my grandparents, it was even more difficult, I didn't dare to ask them [about family disease history]. [. . .] That's why I thought I'd gather my courage for a few more years. (counselee 15)*

Some counselees also expressed reluctance to find out the details of their test result. One counselee expressed feelings of doubt and admitted that he agreed to the test before considering the consequences, but still felt obliged to continue so his daughter would know whether she was at risk. Another counselee said her result was inconclusive, which for her implied that the result was not positive, so it was negative, and that was all she wanted to know about it because she did not want to live her life worrying about genetic risks. One expert confirmed that this occurred regularly.

On the other hand, several counselees gathered a lot of information, stating that this was a helpful strategy for them:

*I had read a lot about BRCA1 all by myself [before genetic counseling] and I actually know quite a lot about it. And I found it [genetic counseling] very exciting, it was more like an educational event, or like a mutual exchange. (counselee 04)*

Proactively gathering information gave those counselees the feeling that they could overcome the inherent complexity. They were eager to learn about their test result and its implications, and it was evident for those counselees that they would take a genetic test.

**Interpreting disease risk.** Counselees interpreted the potential risk of getting cancer in different ways and rated their risk in case of a pathogenic test result. As one expert stated, informing counselees about disease risk was one of the most difficult tasks in genetic counseling. Some counselees and also experts confirmed that risk numbers were interpreted very individually and that for some, quantifying disease risk made a real difference in terms of coping after a variant-positive test result. These counselees reflected on disease probability by making judgments about relative risk numbers that they perceived as "high" or "low" risks to develop a disease.

One counselee, who was skeptical towards genetic testing, reflected that certainty was never absolute and that even a test result that confirmed the presence or absence of a genetic variant could only give "false certainty" (counselee 09). Another counselee stated that even a low cancer risk was rendered useless the moment one actually gets cancer, which for this counselee was the main reason to reject genetic testing.

Some counselees and one expert stressed the difference between carrying a pathogenic genetic variant and actually getting cancer and distinctively considered themselves healthy. This would only change if cancer was actually diagnosed. One counselee stressed that this aspect mainly made him take the genetic test because knowing the risk would allow for preventive measures to stay healthy. For others, in contrast, carrying a pathogenic genetic variant implied disease. One counselee decided against genetic testing because of that perception. Relatedly, another counselee reflected on the anxious feelings he had while waiting for the test result.

In contrast with these considerations, which reflect a certain understanding of disease risk, some counselees restructured risk information to a binary all-or-nothing matter. This was helpful to them to inform their decision-making process:

> I think if I knew I had this mutation, even if the probability is only 40% that I got sick, I would still assume that it happens. (counselee 16).

Relatedly, some counselees reflected on their wish for certainty but understood and acknowledged that this was not realistic in that context:

> So, the question is whether I really want to know that [a genetic disease risk]. Actually, I only want to know that I have zero percent [risk] (laughing). (counselee 13)

Consequently, counselees interpreted their disease risks very individually and these interpretations shaped and informed their decision-making process.

**Using external guidance.** Counselees gave numerous examples of external guidance from family members, health professionals, or health insurance. For some counselees, a major factor was what they thought was best for their children. Experts also confirmed this aspect. Particularly for elderly counselees, this made their decision less complex in the sense that they felt less personally affected:

> I have done this consultation not primarily for me but at my daughter's request, so maybe she was more burdened than I was, I don't know. (counselee 09).

Some counselees heavily relied on their healthcare professionals' recommendations concerning genetic testing. Some of those already affected by cancer considered this recommendation as the main factor affecting their decision, and in some cases, the doctor's recommendation even trumped their own attitude:

> To what extent should you contradict your doctor by saying 'that doesn't make sense'? I'm not enough of a specialist [. . .] to say 'this is a foolish idea'. (counselee 12)

Moreover, two female counselees who were rather skeptical towards genetic testing explained that they followed their trusted gynecologists' advice to go to genetic counseling. It thus depended on the position of the healthcare professional and the attitude of the counselee towards healthcare professionals whether their opinion was used as a heuristic or not.

Some counselees argued that the cost approval by the health insurance facilitated their decision as the medical indication was rechecked and approved. (In Switzerland, health insurance only covers the costs of a genetic test for cancer predisposition if there is a medical indication [41].) So some counselees used the heuristic and gained utility from the fact that their health insurance paid for genetic testing.

Those counselees who decided against genetic testing despite a medical indication did not use such external guidance. Rather, they expressed reluctance to discuss their decision with others who were not part of their closest circle. To them, the decision seemed more complicated because they could not rely on any external guidance. For example, one counselee feared that she would need to justify her decision:

*It's my personal decision and my closest friend knows about it but no one else, that's really my personal thing, yes. [. . .] Possibly because I don't know if I had to justify myself why I didn't do it. (counselee 11)*

Popular individuals such as Angelina Jolie were not perceived as a source of external guidance. Most counselees were aware of Angelina Jolie's public decision to opt for a double mastectomy and a hysterectomy due to a pathogenic *BRCA2* variant [42, 43]. Some said it became easier for them to explain their diagnosis to others because the story had raised awareness among the general population, but none of the counselees reported that Angelina Jolie's example had helped them with their own decision.

**(Re-)Considering the general uncertainty of life.** Some counselees expressed high awareness of the general uncertainty of life, which made them relativize the importance or impact of the genetic test. Some argued that, even if the test was negative, life was still full of (health) risks. They put risk into perspective by illustrating other everyday risks taken despite knowing better, e.g. cycling without a helmet.

*The older I become, the higher the risk that I might have a malfunction in my body, and the health-related risk increases anyway. And it's not necessarily the same disease I already had. It could be something else, just by chance, it could be a lightning strike. And I still feel like having an accident is more likely than having a disease. And that's why I am not scared I could die [from cancer] because there is nothing I can do about it anyway. (counselee 10)*

While those individuals gained little certainty from a genetic test, others felt empowered by knowing and quantitatively assessing their hereditary cancer risk through a genetic test [25].

## Discussion

This study used an inductive approach to identify heuristics used in the genetic testing decision-making process based on qualitative interviews. We found that counselees were very much aware of the complexity inherent to predictive genetic testing decision-making on both a cognitive and an emotional level. We identified six heuristics that counselees used to deal with this complexity. Findings can be aligned with theoretical models of heuristics, including affective, representative, availability, and anchoring heuristics [15]. However, our inductively derived findings indicate that those making the decision intertwine these heuristic types in their individual decision-making process and assign differing practical meaning to them. Our findings provide a deeper understanding of the lived experiences of those going through the genetic testing decision-making process, which can be helpful for healthcare workers providing genetic counseling. Thus, from a practical perspective it is helpful to assess heuristics inductively in a particular context.

Our findings support previous work suggesting that disease risk interpretation has a cognitive as well as an emotional component [44]. Risk interpretation is connected to important uncertainty concerning the probability that a pathogenic genetic test result will actually occur; the ambiguity of whether the pathogenic genetic variant will actually cause disease and whether this could be prevented; and the psychosocial and emotional complexities associated with that assessment [45, 46]. Previous studies have also suggested that decision-makers may use a binary conception of genetic risk as a heuristic for decision-making [32, 47]. Moreover, our finding that social relations (such as respect for (medical) experts or a sense of belonging with family or friends) may be used as external guidance illustrate the wish of some counselees to outsource the decision of whether to take the genetic test to someone else. Other qualitative inquiries about genetic testing decision-making made the same observation [28, 48, 49]. Although it is not per se problematic that people base their medical decisions on relational and social aspects [50], it is important that such external influences are made explicit in genetic counseling.

If we aim for counselees to make an informed decision, meaning that they make their decision incorporating relevant information, these heuristics have differing implications. Table 3 gives a structural overview of our discussion on how these heuristics may influence informed decision-making. We generally grouped them into group A heuristics (helpful to informed decision-making) and group B heuristics (potentially impairing informed decision-making).

On the one hand, the use of heuristics has been widely accepted as a human coping mechanism and are seen as an integral part of decision-making, particularly in complex and uncertain situations [51–53]. Indeed, health psychology research has successfully demonstrated that affective and heuristic components often strongly influence medical decision-making [26, 54]. They can help guide the genetic testing decision-making process [55]. Indeed, group A heuristics (Table 3) were perceived as beneficial by the counselees in our study as they helped them to operationalize information to make their decision, which is useful to be encouraged in genetic counseling [56]. They all are based on simple decision trees with yes/no-options: "If XY happens, I will do Z", which is based on the heuristics of factual knowledge [16]. Consequently, complex information became manageable on a cognitive level and perceived uncertainty decreased on an emotional level, resulting in a logical sequence of actions. Experts also confirmed that they supported this heuristic by proactively discussing potential implications

**Table 3. Potential influence of observed heuristics on informed decision-making.**

| Group | Observed heuristics | Assessment regarding informed decision-making |
|---|---|---|
| A | Anticipating the test result | Usually assisted by a pedigree analysis which serves as an external mapping of intuitive feelings–> no harm |
| A | Focusing on consequences | Experience of self-efficiency |
| A | Dealing with information | In our sample, not harmful; counselees returned to the topic if they wanted to. Important to have information available that people can rely on when they are ready |
| B | Interpreting disease risk | In general a common and useful heuristic, but in combination with strong emotions it could be potentially harmful to informed decision-making |
| B | Using external guidance | Can assist the coping mechanism, not harmful as long as family members and healthcare professionals are aware of their role and do not try to manipulate the decision |
| B | (Re-)Considering the general uncertainty of life | Might be helpful to avoid being stuck in long recurrent reflections, on the other hand, health care and prevention do have a positive impact on health status/life span, and therefore leaving everything to chance might cause serious harm |

and consequences with counselees in genetic counseling sessions. Counselees who avoided information about genetic testing at some point also tended to apply this kind of if-then logic: For example, the rationale to seek more information and do the genetic test once it is more easily accessible. Genetic counseling encourages this kind of reasoning by illustrating the consequences of a negative or positive result of the genetic testing and the respective preventive options [57].

On the other hand, while group B heuristics (Table 3) also seem to facilitate decision making but also have potentially harmful aspects in the sense that they might hinder informed decision-making or foster anxiety and other psychosocial problems. Therefore, these heuristics should receive special attention in genetic counseling. Group B heuristics tend to trigger strong emotions, such as absolute euphoria or profound unhappiness in case of a negative or positive result in genetic testing. These strong emotions might complicate the decision-making process and cause psychosocial harm. They might hinder individuals to make an informed decision because, in the presence of strong emotions, it is also more complicated to process information, particularly in uncertain situations [58]. For instance, the (re-)consideration of the uncertainty of human life is the counterexample to the above-mentioned decision tree heuristic. Instead of logical decision sequences, such reasoning might become circular, with no escape. This might produce stress, a feeling of losing control, and desperation. On the other hand, such considerations of inherent life uncertainty reflect counselees' general life philosophy, as our group found in a different analysis based on the same data [25]. Consequently, our findings underline how vital it is for informed decision making that genetic counselors attempt to understand the values and life philosophy of their individual counselees and address them, as those preferring to leave life to fate might be more prone to enter harmful cycles of brooding. Nonetheless, when trying to resolve misunderstandings of facts, genetic counselors still generally have the ethical and legal duty to respect their counselees' decisions even if they appear irrational, but they may point out hasty or irrational decision making. They may even legitimately refuse to prescribe genetic testing if providing it would be unethical, and there is also an absolute negative right of the counselee to refuse testing even if the geneticist finds that it is indicated [59].

In our sample, the heuristic of avoiding information was not perceived or interpreted as impairing informed decision-making, as it mainly concerned young individuals who opted for genetic testing later in life. Others deliberately chose not to learn more about certain particular implications of their genetic test, which to them was more important than knowing the details. Yet, information avoidance has been reported as problematic to informed decision-making in the genetic counseling literature [60]. Because visiting at least one genetic counseling session was an inclusion criterion to participate in this study, none of the counselees avoided the topic completely, which might be the reason why it was not considered problematic in our study.

Our findings are limited to the decision-making context of predictive genetic testing for hereditary cancer syndromes. Our counselees' situations were characterized by incomplete disease penetrance (meaning that carrying a pathogenic genetic variant does not necessarily lead to disease) and conducted genetic tests were generally limited to target-specific or panel analyses [23]. However, other circumstances of genetic testing might lead to different complexities and, therefore, to differing use of heuristics.

This was a qualitative study involving 18 counselees and two experts. We experienced difficulty in recruiting enough patients and had to stop recruitment after two and a half years for reasons of time constraints. (This did not apply to experts where we decided to stop recruitment based on theoretical saturation.) When assessing data saturation for themes during the data analysis process, we found the themes identified well saturated and confirmed by expert interviews. Still, the demographic distribution of the sample might point to potential

shortcomings in terms of theoretical saturation. First, we had interviewed 14 women and only four men. Second, most participants considered testing for hereditary breast/ovarian cancer syndrome, which is comparatively well-known among the general public [61, 62] and well established in the clinical setting [63]. These factors might influence decision-making as compared to less established hereditary cancer syndromes. Third, because counselees were recruited through counseling physicians, those deciding against genetic testing were difficult to recruit and are underrepresented (only three of the interviewed counselees decided against testing). Those counselees who completely refuse genetic testing might not even bother seeing a genetic counselor and are thus not included in this study, even though from a conceptual perspective these views might have been an interesting addition to our findings. However, this does not affect the practical relevance of this study, as findings include views of those seeing a genetic counselor.

## Conclusion

This study confirms that heuristics are a natural part of human decisions and are not opposed to informed decision making by those going through the decision making process of genetic testing for inherited cancer risks. On the contrary, heuristics are crucial in guiding counselees through the complexities of genetic testing decision-making. However, some heuristics can increase the risk of misinterpretation or exaggerated external influences. As this may impair informed decision making, it is important to adequately address heuristics in genetic counseling. Our findings further imply that strong emotions (both positive and negative) require special attention as they might lead to hasty decisions as a result of plain patterns. They might also hinder successful information transfer. To address these emotions along genetic counseling is important. It will be helpful for informed decision-making if the identified heuristics are addressed in genetic counseling to maximize their benefits and minimize the potential impairment some of them might cause. As such, the study underlines the importance of patient-centered, personalized genetic counseling that has the resources to address patient-specific heuristics and interpretations.

## Supporting information

**S1 File. Author details and COREQ checklist.**
(DOCX)

**S2 File. Interview guide patient interviews.**
(DOCX)

**S3 File. Interview guide expert interviews.**
(DOCX)

**S4 File. Coding framework.**
(DOCX)

## Acknowledgments

We thank the four counseling physicians who recruited counselees for this study.

## Author Contributions

**Conceptualization:** Bettina Maria Zimmermann, David Martin Shaw, Bernice Elger.

**Data curation:** Bettina Maria Zimmermann.

**Formal analysis:** Insa Koné.

**Funding acquisition:** Bettina Maria Zimmermann.

**Investigation:** Bettina Maria Zimmermann.

**Methodology:** Bettina Maria Zimmermann.

**Project administration:** Bettina Maria Zimmermann.

**Resources:** Bernice Elger.

**Supervision:** David Martin Shaw, Bernice Elger.

**Validation:** David Martin Shaw, Bernice Elger, Insa Koné.

**Visualization:** Insa Koné.

**Writing – original draft:** Bettina Maria Zimmermann, Insa Koné.

**Writing – review & editing:** Bettina Maria Zimmermann, David Martin Shaw, Bernice Elger, Insa Koné.

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
