## [Decision Letter · Decision Letter 0]

19 May 2021

PONE-D-21-09002

Decision making in genetic testing for hereditary cancer: Patients’ approaches to complexity

PLOS ONE

Dear Dr. Zimmermann,

Thank you for submitting your manuscript to PLOS ONE. After careful consideration, we feel that it has merit but does not fully meet PLOS ONE’s publication criteria as it currently stands. Therefore, we invite you to submit a revised version of the manuscript that addresses the points raised during the review process.

We look forward to receiving your revised manuscript.

Kind regards,

Prof. Ritesh G. Menezes, M.B.B.S., M.D., Diplomate N.B.

Academic Editor

PLOS ONE

Journal Requirements:

Reviewers' comments:

Reviewer's Responses to Questions

**Comments to the Author**

1. Is the manuscript technically sound, and do the data support the conclusions?

Reviewer #1: Partly

Reviewer #2: Yes

Reviewer #3: Yes

Reviewer #4: Yes

2. Has the statistical analysis been performed appropriately and rigorously? 

Reviewer #1: N/A

Reviewer #2: N/A

Reviewer #3: N/A

Reviewer #4: N/A

3. Have the authors made all data underlying the findings in their manuscript fully available?

Reviewer #1: No

Reviewer #2: No

Reviewer #3: No

Reviewer #4: No

4. Is the manuscript presented in an intelligible fashion and written in standard English?

Reviewer #1: Yes

Reviewer #2: Yes

Reviewer #3: Yes

Reviewer #4: Yes

5. Review Comments to the Author

Reviewer #1: While this research contributes to a more comprehensive understanding of decision-making about genetic testing in the context of hereditary cancer, the manuscript would benefit from the following:

1. A description of the study sample, including the variability that is represented in the participants. The rational for recruiting 2 physicians was missing and would perhaps be important as much of the introduction and discussion focusses on the role of genetic counsellors.

2. Further development of the findings. While the authors present a theoretical structure based on identified heuristics, it was not necessarily clear the ways in which the identified heuristics influenced decision making in an in-depth manner. Additional detail would strengthen the findings.

3. Categorization of heuristics as unproblematic vs problematic. The description of the heuristics as unproblematic vs problematic in the discussion appears to be the authors interpretations and more fitting in the findings section. However, it was not clear what the authors meant by unproblematic vs problematic - for whom? problematic in what way? for what end goal - patient coping and adjustment, making of an informed decision, making a decision that aligns with values and beliefs, making a decision that mitigates family conflict?? There seems to be inherent assumptions about what is a "good" versus a "bad" decision making process that if made explicit would provide greater clarity of the authors arguments. I also found that determination of the heuristics as either unproblematic or problematic was not sufficiently supported by the presented data or findings and that the binary was not necessarily support by the description in the 3rd column of Tables 1 & 2.

4. The authors state (page 19, line 393) that"the results are still valid," This language about validity in the context of grounded theory/qualitative research requires clarification and perhaps revision depending on the authors conceptualization and use of the term valid here.

Reviewer #2: Overview:

The overall thematic of heuristics to evaluate the decision-making process in genetic-testing for hereditary cancer is respected. The paper shows the perspective of those who have to make the decision to take the genetic-test or not, and to cope with the decision they made and the results. In situations of uncertainty and complexity, the relationship between at-risk persons and health professionals are important. This qualitative research brings to light the perspective of those directly concerned.

However, there are some points to be changed or improved. First two major changes: add the demographic table and complete the section limitations accordingly. Second, don’t use “should”. The other points mentioned in the detailed review are minor.

NB: I did not have access to the 2 supporting information files.

Detailed review:

1. The title speak about patients, but not all participants were patients. In the demographics table disclosed in the initial study (reference 25) there are 6 participants with cancer and 5 with a known mutation in the family. It means that 7 to 12 participants may not be patients per se, but only candidates for genetic-testing.

2. Introduction and methodology

The study is a second analysis of qualitative data, which is possible in qualitative research in order to develop more in-depth analysis. In this secondary analysis, there is an intention to deepen the analysis based on an initial framework of heuristics. It is a thematic approach based on a modified approach to grounded theory and not a typical grounded theory methodology.

The reference 18 (Peters 2006) is cited identifying four heuristics in the introduction. I will recommend to also cite the definition of heuristics from this paper “the mental shortcuts or“rules of thumb” that decision makers consciously or unconsciously employ to make judgments of uncertainty” as heuristics is one of the main research question in the final paragraph of the introduction and it is also the aim of the data analysis in the section methodology.I will also recommend to be more coherent in the declared aim of the research and the aim of data analysis between introduction and methodology.

3. Findings

a. Demographics are missing. You need to cite or reproduce the table from the initial published study (reference 25). It is a very relevant table as we can see important data for the discussion and the limitations of your study.

b. Experiencing complexity. This first result is important as it gives the complexity of the situation: emotion, family context, uncertainty…The issue of language with “mutation” versus “variant” could be more developed (if the data allow it) as it has a direct implication on how medical doctors /genetic counsellor could speak.

c. Six heuristics themes are identified.

This part is the result of interpretation and complex analysis that I cannot assess precisely without doing again the whole analysis, which is not the purpose of this review.

I will recommend adapting the title “avoiding information” as this section speaks also about some participants who gather a lot of information. Moreover, those who postpone the decision may do it, not because they want to avoid information, but because they wait to reach the age when a medical prescription will be made which means in Switzerland the age when the social insurance will cover the cost. Maybe the title could be “dealing with information” or something like that. In the table 1 of the discussion, this could be changed without modifying the sense.

In the section “using external guidance” you mention this need for a medical indication to cover the cost. But in the recruitment part of the methodology, it is written lines 121-122 that : “They all had a medical indication for genetic testing according to Swiss or international medical guidelines, particularly cancer diagnosis at an early age or important family history”. It means that retrospectively some participants who took the test were influenced by the cost, but there are also 3 patients with medical indications who did not take the test in the sample. It seems that the external guidance based on cost might be only relevant to the young ones postponing the test.

The title “(Re-)Considering the general uncertainty of life” is referring to the results of the first paper (ref.25). You could mention it there, even if it is then well described in discussion.

4. Discussion

It is always difficult to draw normative conclusions from empirical data. The presentation in tables has the merit to organize the reflection.

The recommendation to genetic counsellors to take into account the values and emotions of patients, the external influencing factors, seems reasonable. There is nothing new, but the study has the advantage to confirm it from the perspective of those who have to make the decision. It brings sense to what genetic counsellors are doing.

However: Avoid “should” in the abstract line 40, in the discussion lines 371, 373 374, and in the conclusion line 405. This research paper has not the authority to command a “should”. Rather try to formulate this with some sentences showing how important it is for medical doctors prescribing genetic-testing, and for genetic counsellors to …….”. If the genetic counsellor is not prepared to listen to values and context of the at-risk person, the Swiss law will not make compulsory to have a consultation with her/him before deciding on genetic-testing.

5. Limitations:

The gender bias has to be mentioned in the limitations. There are 14 women and only 4 men. Most of the participants (n=16) have an history of breast / ovarian cancer. Without any discriminating approach, it is well known that there is a gender effect in the balance cognitive/emotional reaction. Breast cancer is also the most frequent cancer in women and is intensively publicly mentioned (for instance campaign for mammography). This could also explain that few recruited participants decided not to take the test.

Reviewer #3: In this manuscript, Zimmermann et al. use a grounded theory approach to explore the heuristics used by participants during decision making to undergo predictive genetic testing for hereditary cancer syndromes. They identify 6 heuristics used by participants to help simplify their decisions; and they further divide these 6 heuristics as either problematic or unproblematic, depending on whether they did or did not have aspects that the authors judged to be potentially harmful or create anxiety or psychosocial concerns.

The paper is well written and addresses an important clinical topic, namely participant decision making for predictive testing for cancer risk. By extension, the heuristics identified are useful for those providing counseling about such testing.

It is not clear however, why the authors chose to include 2 counseling physicians in their interviews; and furthermore, having decided to include counseling physicians, why they included only 2 of them, while there are 18 participants who were deciding whether to undergo testing or not. Based on the Methods described, it appears that the physicians were interviewed after the interviews of the patients and the physician interviews asked them about “…issues that emerged during analysis.” It isn't clear to me whether they had already developed their heuristics prior to physician interviews, or whether there was anything of patient responses shared with the physicians. This information would be valuable to include in their Methods.

Subsequent to the Methods section, there is no mention of the physician interviews, and the heuristics and the remainder of the manuscript focuses only on the patient responses.

So, from a methodological standpoint, it would be helpful for the authors to describe the rationale for inclusion of physicians, since it appears that the physician responses were not used to identify the heuristics. If on the other hand, physician responses were in some way helpful for identification of heuristics or for other issues such as validating the complexity of such counseling, then this should be included in both Results and Discussion. And, from a scientific standpoint, it would be helpful for the authors to report on any differences and similarities between findings in the patient and physician groups. On the other hand, if the heuristics identified are all the result of analysis of patients’ interviews, perhaps inclusion of physician interviews is not pertinent to the overall results of the analysis and don’t merit inclusion.

Reviewer #4: The authors have investigated an important and timely research topic with clear practice implications. I have made suggestions to improve the transparency and structure of this manuscript.

General comments

1. I suggest using a qualitative reporting checklist like the COREQ guidelines to enhance the transparency of methods. [Tong A, Sainsbury P, Craig J. Consolidated criteria for reporting qualitative research (COREQ): a 32-item checklist for interviews and focus groups. International journal for quality in health care. 2007 Dec 1;19(6):349-57.]

Introduction

2. It would be helpful to include a clear discussion of the evidence gap that that authors have intended to address. What has been done on this topic before and what is the novel contribution of this work? Separate paragraphs to address the evidence gap and the specific objectives of this study would improve the introductory section.

Methods

3. Can the authors please explain the context of the “larger grounded theory study” that they refer to in the first sentence?

4. It is unclear why the authors interviewed 18 patient participants and only 2 physicians. How was the decision made to interview only 2 physicians? Similarly, why did recruitment stop at 18 patients? There is no mention of thematic saturation included in the methods description.

5. Why were pregnant people excluded from interviews?

6. Please explain how the interview guide was developed. Was literature review or pilot testing involved? How were the hypothetical vignettes developed? Was the guide revised throughout the interview process?

7. Were the physician interviews analyzed along with the patient (at-risk) participants, or separately?

8. Did the authors collect demographic information from participants? E.g. age/ gender etc? If so, please report.

9. I am curious as to why research participants are referred to as “at risk individuals” throughout the paper. Switching between different terms such as “participants” and “at risk individuals” is unclear.

Results

10. The beginning of the results section would benefit from a high-level summary of the major themes/ sub-themes generated through the analysis, as a preface to the more detailed description.

11. As above, this section would benefit from a description of the characteristics of the participants, if this information was collected.

12. Could the authors please attach the final coding framework as an additional supplemental document? Attaching and linking the results section to the framework may help to provide a clearer structure to this entire section.

13. Within the results section, can the authors speak to any of these themes where participants were generally consistent in their opinions, versus where opinions varied? This would also help to frame the discussion around practice implications.

14. For ease of reading, I suggest pulling all direct quotes out of the paragraphs and including in indented, italicised font.

15. The link between the patient participants and the 2 physician participants remains unclear. The authors may consider reporting their results separately, given that two separate interview guides were used, and only 2 physicians were included.

Discussion

16. I suggest placing tables 1 and 2 and the associated discussion into the results section

17. This section would benefit from a more comprehensive discussion of how findings deviate or align with existing work related to values and preferences related to hereditary cancer genetic testing. Specifically (and as mentioned in introduction comments) what is novel about this contribution to the evidence base and how can findings be used to inform practice/ policy/ future research?

18. The following sentence requires clarification/ elaboration, “Because our results should inform genetic counseling practice, our results are still valid, even though from a conceptual perspective the view of those even refusing to go to genetic counseling would have been highly interesting.”

19. Pending elaboration on recruitment methods, the authors may need to include additional limitations related to their sampling and cessation of recruitment (e.g. why were 18 interviews conducted with 18 at risk individuals and only 2 with physicians)

6. PLOS authors have the option to publish the peer review history of their article (what does this mean?). If published, this will include your full peer review and any attached files.

Reviewer #1: No

Reviewer #2: No

Reviewer #3: No

Reviewer #4: No

---

## [Author Response · Author response to Decision Letter 0]

15 Jun 2021

Journal Requirements:

RESPONSE: We have adapted the manuscript to the journal's style requirements.

2. In your Data Availability statement, you have not specified where the minimal data set underlying the results described in your manuscript can be found. PLOS defines a study's minimal data set as the underlying data used to reach the conclusions drawn in the manuscript and any additional data required to replicate the reported study findings in their entirety. All PLOS journals require that the minimal data set be made fully available. For more information about our data policy, please see http://journals.plos.org/plosone/s/data-availability. Upon re-submitting your revised manuscript, please upload your study’s minimal underlying data set as either Supporting Information files or to a stable, public repository and include the relevant URLs, DOIs, or accession numbers within your revised cover letter. For a list of acceptable repositories, please see http://journals.plos.org/plosone/s/data-availability#loc-recommended-repositories. Any potentially identifying patient information must be fully anonymized. Important: If there are ethical or legal restrictions to sharing your data publicly, please explain these restrictions in detail. Please see our guidelines for more information on what we consider unacceptable restrictions to publicly sharing data: http://journals.plos.org/plosone/s/data-availability#loc-unacceptable-data-access-restrictions. Note that it is not acceptable for the authors to be the sole named individuals responsible for ensuring data access. We will update your Data Availability statement to reflect the information you provide in your cover letter.

RESPONSE: Based on the Swiss Human Research Act (HRA) and the ethics approval obtained for this study, we are legally bound to keep the identity of study participants confidential. Due to the highly identifiable topic of this research (personal and family disease history for relatively rare diseases – hereditary cancer syndromes – and the regionally limited recruitment method, interview transcripts cannot be fully anonymous. We do provide anonymized quotes within the manuscript to illustrate our findings (corresponding to transcript excerpts) and thus confirm with the journal's "guidelines for qualitative data".

Reviewer #1

While this research contributes to a more comprehensive understanding of decision-making about genetic testing in the context of hereditary cancer, the manuscript would benefit from the following:

RESPONSE: We are grateful for reviewer 1's feedback on our study and will reply to their comments in detail below.

Methods

1. A description of the study sample, including the variability that is represented in the participants. The rational for recruiting 2 physicians was missing and would perhaps be important as much of the introduction and discussion focusses on the role of genetic counsellors.

RESPONSE: We have included the demographics of counselees as Table 1 (see results section). We also provide now a rationale for recruiting two experts in the methods section. It now reads: "Expert interviews were conducted after initial analysis of all counselee interviews and included questions about issues that emerged during analysis, including questions on uncertainty and risk perception (see Supporting information file S2). The purpose of expert interviews was to add the experts' perspective to the analysis and to compare preliminary findings with expert knowledge. Because no new relevant aspects came up, data collection of expert interviews was completed after two interviews. "

Results

2. Further development of the findings. While the authors present a theoretical structure based on identified heuristics, it was not necessarily clear the ways in which the identified heuristics influenced decision making in an in-depth manner. Additional detail would strengthen the findings.

RESPONSE: We have included more explicit references in the findings on how heuristics influenced decision-making. An additional overview is given in Tables 2 and 3.

Discussion

3. Categorization of heuristics as unproblematic vs problematic. The description of the heuristics as unproblematic vs problematic in the discussion appears to be the authors interpretations and more fitting in the findings section. However, it was not clear what the authors meant by unproblematic vs problematic - for whom? problematic in what way? for what end goal - patient coping and adjustment, making of an informed decision, making a decision that aligns with values and beliefs, making a decision that mitigates family conflict?? There seems to be inherent assumptions about what is a "good" versus a "bad" decision making process that if made explicit would provide greater clarity of the authors arguments. I also found that determination of the heuristics as either unproblematic or problematic was not sufficiently supported by the presented data or findings and that the binary was not necessarily support by the description in the 3rd column of Tables 1 & 2.

RESPONSE: Thank you, we have rephrased this to be more clear on the context of informed decision-making. Indeed, implications on informed decision-making was not based on the data directly but our interpretation of the findings. We therefore find it more suitable to separate those from what was actually grounded in interview data and kept this part in the discussion. We made this more transparent in the paper.

4. The authors state (page 19, line 393) that "the results are still valid," This language about validity in the context of grounded theory/qualitative research requires clarification and perhaps revision depending on the authors conceptualization and use of the term valid here.

RESPONSE: Thank you, we agree with this reviewer that the term "valid" would need further explanation. We decided that it is easier to rephrase the meaning of our argument, avoiding this term. It now reads: "Those counselees who completely refuse genetic testing might not even bother seeing a genetic counselor and are thus not included in this study, even though from a conceptual perspective these views might have been an interesting addition to our findings from a conceptual perspective. Yet, this does not affect the practical relevance of this study."

Reviewer #2

Overview: The overall thematic of heuristics to evaluate the decision-making process in genetic-testing for hereditary cancer is respected. The paper shows the perspective of those who have to make the decision to take the genetic-test or not, and to cope with the decision they made and the results. In situations of uncertainty and complexity, the relationship between at-risk persons and health professionals are important. This qualitative research brings to light the perspective of those directly concerned.

However, there are some points to be changed or improved. First two major changes: add the demographic table and complete the section limitations accordingly. Second, don’t use “should”. The other points mentioned in the detailed review are minor.

NB: I did not have access to the 2 supporting information files.

RESPONSE: We thank reviewer 2 for their helpful comments. We have addressed this reviewer's suggestions and requests and reply to them in detail below.

Detailed review:

1. The title speak about patients, but not all participants were patients. In the demographics table disclosed in the initial study (reference 25) there are 6 participants with cancer and 5 with a known mutation in the family. It means that 7 to 12 participants may not be patients per se, but only candidates for genetic-testing.

RESPONSE: Thank you, we have addressed this issue by revising the title and, in the manuscript, by consistently referring to "counselees" when talking about those making the decision for or against genetic counseling, "experts" for genetic counseling physicians, and "participants" when referring to all of them together (experts + counselees). 

2. Introduction and methodology

The study is a second analysis of qualitative data, which is possible in qualitative research in order to develop more in-depth analysis. In this secondary analysis, there is an intention to deepen the analysis based on an initial framework of heuristics. It is a thematic approach based on a modified approach to grounded theory and not a typical grounded theory methodology.

RESPONSE: Thank you, we have acknowledged this comment in the "data analysis" subsection.

The reference 18 (Peters 2006) is cited identifying four heuristics in the introduction. I will recommend to also cite the definition of heuristics from this paper “the mental shortcuts or“rules of thumb” that decision makers consciously or unconsciously employ to make judgments of uncertainty” as heuristics is one of the main research question in the final paragraph of the introduction and it is also the aim of the data analysis in the section methodology.I will also recommend to be more coherent in the declared aim of the research and the aim of data analysis between introduction and methodology.

RESPONSE: Thank you, we followed this suggestion and introduced the definition as used by Peters et al 2006.

3. Findings

a. Demographics are missing. You need to cite or reproduce the table from the initial published study (reference 25). It is a very relevant table as we can see important data for the discussion and the limitations of your study.

RESPONSE: Thank you, we have included the participant demographics as Table 1.

b. Experiencing complexity. This first result is important as it gives the complexity of the situation: emotion, family context, uncertainty…The issue of language with “mutation” versus “variant” could be more developed (if the data allow it) as it has a direct implication on how medical doctors /genetic counsellor could speak.

RESPONSE: Thank you, we added an additional sentence on that topic. It reads: "Conversely, one expert reflected on the difficulty of explaining these variants to counselees in the context of panel testing, stating that those lower risks or variants of unknown significance added an additional layer of complexity that was difficult to communicate. The expert rather feared that counselees would overestimate the risks of these variants."

c. Six heuristics themes are identified.

This part is the result of interpretation and complex analysis that I cannot assess precisely without doing again the whole analysis, which is not the purpose of this review.

I will recommend adapting the title “avoiding information” as this section speaks also about some participants who gather a lot of information. Moreover, those who postpone the decision may do it, not because they want to avoid information, but because they wait to reach the age when a medical prescription will be made which means in Switzerland the age when the social insurance will cover the cost. Maybe the title could be “dealing with information” or something like that. In the table 1 of the discussion, this could be changed without modifying the sense.

RESPONSE: Thank you, we adapted as suggested.

In the section “using external guidance” you mention this need for a medical indication to cover the cost. But in the recruitment part of the methodology, it is written lines 121-122 that : “They all had a medical indication for genetic testing according to Swiss or international medical guidelines, particularly cancer diagnosis at an early age or important family history”. It means that retrospectively some participants who took the test were influenced by the cost, but there are also 3 patients with medical indications who did not take the test in the sample. It seems that the external guidance based on cost might be only relevant to the young ones postponing the test.

RESPONSE: We think that there is a misunderstanding that we attempted to avoid by reformulating the part on health insurance coverage: participants were not influenced by cost but the external reference point of the health insurance's decision to cover their test gave them an additional reason to take the test (health insurance wouldn't cover it if test was useless). 

The title “(Re-)Considering the general uncertainty of life” is referring to the results of the first paper (ref.25). You could mention it there, even if it is then well described in discussion.

RESPONSE: Thank you, we have added the reference as suggested.

4. Discussion

It is always difficult to draw normative conclusions from empirical data. The presentation in tables has the merit to organize the reflection. 

RESPONSE:

The recommendation to genetic counsellors to take into account the values and emotions of patients, the external influencing factors, seems reasonable. There is nothing new, but the study has the advantage to confirm it from the perspective of those who have to make the decision. It brings sense to what genetic counsellors are doing. However: Avoid “should” in the abstract line 40, in the discussion lines 371, 373 374, and in the conclusion line 405. This research paper has not the authority to command a “should”. Rather try to formulate this with some sentences showing how important it is for medical doctors prescribing genetic-testing, and for genetic counsellors to …….”. If the genetic counsellor is not prepared to listen to values and context of the at-risk person, the Swiss law will not make compulsory to have a consultation with her/him before deciding on genetic-testing.

RESPONSE: Thank you, we have reformulated the sections mentioned, avoiding the term "should".

5. Limitations:

The gender bias has to be mentioned in the limitations. There are 14 women and only 4 men. Most of the participants (n=16) have an history of breast / ovarian cancer. Without any discriminating approach, it is well known that there is a gender effect in the balance cognitive/emotional reaction. Breast cancer is also the most frequent cancer in women and is intensively publicly mentioned (for instance campaign for mammography). This could also explain that few recruited participants decided not to take the test.

RESPONSE: Thank you, we have acknowledged this in the limitations section.

Reviewer #3

In this manuscript, Zimmermann et al. use a grounded theory approach to explore the heuristics used by participants during decision making to undergo predictive genetic testing for hereditary cancer syndromes. They identify 6 heuristics used by participants to help simplify their decisions; and they further divide these 6 heuristics as either problematic or unproblematic, depending on whether they did or did not have aspects that the authors judged to be potentially harmful or create anxiety or psychosocial concerns.

The paper is well written and addresses an important clinical topic, namely participant decision making for predictive testing for cancer risk. By extension, the heuristics identified are useful for those providing counseling about such testing.

It is not clear however, why the authors chose to include 2 counseling physicians in their interviews; and furthermore, having decided to include counseling physicians, why they included only 2 of them, while there are 18 participants who were deciding whether to undergo testing or not. Based on the Methods described, it appears that the physicians were interviewed after the interviews of the patients and the physician interviews asked them about “…issues that emerged during analysis.” It isn't clear to me whether they had already developed their heuristics prior to physician interviews, or whether there was anything of patient responses shared with the physicians. This information would be valuable to include in their Methods.

Subsequent to the Methods section, there is no mention of the physician interviews, and the heuristics and the remainder of the manuscript focuses only on the patient responses. So, from a methodological standpoint, it would be helpful for the authors to describe the rationale for inclusion of physicians, since it appears that the physician responses were not used to identify the heuristics. If on the other hand, physician responses were in some way helpful for identification of heuristics or for other issues such as validating the complexity of such counseling, then this should be included in both Results and Discussion. And, from a scientific standpoint, it would be helpful for the authors to report on any differences and similarities between findings in the patient and physician groups. On the other hand, if the heuristics identified are all the result of analysis of patients’ interviews, perhaps inclusion of physician interviews is not pertinent to the overall results of the analysis and don’t merit inclusion.

RESPONSE: We thank reviewer 3 for their insightful comments and their positive assessment of our study. The rationale to include expert interviews was to verify and contrast our findings with the perspective of genetic counseling experts. We conducted them after initial analysis of all patient interviews and included in the interview guide the aspects we deemed most relevant for our findings. Relevant for this paper, we included questions concerning uncertainty and risk perception. We have clarified the role of expert interviews in the methods section, it now reads: "Expert interviews were conducted after initial analysis of all counselee interviews and included questions about issues that emerged during analysis, including questions on uncertainty and risk perception (see Supporting information file S2). The purpose of expert interviews was to add the experts' perspective to the analysis and to compare preliminary findings with expert knowledge. Because no new relevant aspects came up, data collection of expert interviews was completed after two interviews." We also implemented the expert interviews more visibly in the findings, stating in what instances their perspectives were in line or contrasting views of counselees.

Reviewer #4

The authors have investigated an important and timely research topic with clear practice implications. I have made suggestions to improve the transparency and structure of this manuscript.

RESPONSE: We thank reviewer 4 for their extensive and insightful comments that indeed were very supportive to improve the paper from our perspective. We reply to this reviewer's comments in detail below.

General comments

1. I suggest using a qualitative reporting checklist like the COREQ guidelines to enhance the transparency of methods. [Tong A, Sainsbury P, Craig J. Consolidated criteria for reporting qualitative research (COREQ): a 32-item checklist for interviews and focus groups. International journal for quality in health care. 2007 Dec 1;19(6):349-57.]

RESPONSE: We have consulted and report on the COREQ checklist in Supporting Information File S1.

Introduction

2. It would be helpful to include a clear discussion of the evidence gap that that authors have intended to address. What has been done on this topic before and what is the novel contribution of this work? Separate paragraphs to address the evidence gap and the specific objectives of this study would improve the introductory section.

RESPONSE: Thank you, we have introduced a separate paragraph after presenting the aims of the study that includes a small literature review to illustrate the research gap that we intend to fill, which reads: "By contrast to previous inquiries on the role of heuristics in the genetic testing decision-making process, our study takes an inductive approach to investigate the influence of heuristics and focusses on the implications for informed decision-making in the context of genetic testing for inherited cancer risk."

Methods

3. Can the authors please explain the context of the “larger grounded theory study” that they refer to in the first sentence?

RESPONSE: We have added some more contextual information.

4. It is unclear why the authors interviewed 18 patient participants and only 2 physicians. How was the decision made to interview only 2 physicians? Similarly, why did recruitment stop at 18 patients? There is no mention of thematic saturation included in the methods description.

RESPONSE: We added more information on the rationale for recruitment, why it stopped and our approach to data saturation in the methods section. It now reads: "During the recruitment process, we found that the views of males, healthy individuals, and those deciding against genetic testing were underrepresented yet potentially important for theoretical saturation. We therefore purposefully recruited for these participants in a second part of the study. Yet, we had difficulties recruiting enough patients and had to stop recruitment after twoandahalf years for reasons of time constraints. We applied a pragmatic approach to data saturation and critically examined saturation of themes during the data analysis process [28]."

We also added information on the role and rationale behind the expert interviews: "Expert interviews were conducted after initial analysis of all counselee interviews and included questions about issues that emerged during analysis, including questions on uncertainty and risk perception (see Supporting information file S2). The purpose of expert interviews was to add the experts' perspective to the analysis and to compare preliminary findings with expert knowledge. Because no new relevant aspects came up, data collection of expert interviews was completed after two interviews."

5. Why were pregnant people excluded from interviews?

RESPONSE: This was purely for organizational reasons; including pregnant women would have rendered the study to a higher risk-level in terms of ethics approval. We have rephrased to make this more clear in the methods section, it now reads: "Counsellees between the age of 18-70 that were not considered vulnerable due to mental state or pregnancy were eligible."

6. Please explain how the interview guide was developed. Was literature review or pilot testing involved? How were the hypothetical vignettes developed? Was the guide revised throughout the interview process?

RESPONSE: We have added additional explanations in the methods section. It reads: "The interview guide was pilot tested with two healthy volunteers (not included in this study) and adjusted after the six interviews and again after eleven interviews based on preliminary findings (see Supporting information file S1 for the interview guide). We also presented three hypothetical scenarios of genetic testing decision-making and asked them how they would decide in these scenarios, and why. Scenarios were developed to address and enrich counselees' reflections on risk perception and medical actionability (scenario 1), relative importance of genetic testing in counselees' lives (scenario 2), and environmental influence on disease outbreak (scenario 3)."

7. Were the physician interviews analyzed along with the patient (at-risk) participants, or separately?

RESPONSE: We have specified accordingly, it reads: " With the expert interviews, we also sought to identify potential additional aspects and alternative explanations. They were analysed along with the other interviews."

8. Did the authors collect demographic information from participants? E.g. age/ gender etc? If so, please report.

RESPONSE: Thank you, we have included the participant demographics as Table 1.

9. I am curious as to why research participants are referred to as “at risk individuals” throughout the paper. Switching between different terms such as “participants” and “at risk individuals” is unclear.

RESPONSE: Thank you, we attempted to make this more clear by consistently referring to "counselees" when talking about those making the decision for or against genetic counseling, "experts" for genetic counseling physicians, and "participants" when referring to all of them together (experts + counselees). 

Results

10. The beginning of the results section would benefit from a high-level summary of the major themes/ sub-themes generated through the analysis, asc a preface to the more detailed description.

RESPONSE: We have implemented a short introductory paragraph at the beginning of the findings. 

11. As above, this section would benefit from a description of the characteristics of the participants, if this information was collected.

RESPONSE: Thank you, we have included the participant demographics as Table 1.

12. Could the authors please attach the final coding framework as an additional supplemental document? Attaching and linking the results section to the framework may help to provide a clearer structure to this entire section.

RESPONSE: We have included the coding framework as supplementary file 3.

13. Within the results section, can the authors speak to any of these themes where participants were generally consistent in their opinions, versus where opinions varied? This would also help to frame the discussion around practice implications.

RESPONSE: Generally, we found highly individual and variable experiences which we attempted to outline in our findings. We purposefully refrain from making any quantitative claims (in the sense that some experiences were more often mentioned than others) to avoid readers to make relative quantitative conclusions (such as some experiences are more common than others). Due to low sample size, this is not an interpretation we can take from this study.

14. For ease of reading, I suggest pulling all direct quotes out of the paragraphs and including in indented, italicised font.

RESPONSE: We formatted the results section as suggested.

15. The link between the patient participants and the 2 physician participants remains unclear. The authors may consider reporting their results separately, given that two separate interview guides were used, and only 2 physicians were included.

RESPONSE: The purpose for the expert interviews was to add an additional perspective and verify our findings. That is also why only two expert interviews were conducted; they did not reveal any new insights and were seen as an indication of theoretical saturation. We have included a number of direct references where experts confirmed or contrasted the views of counselees in the results section.

Discussion

16. I suggest placing tables 1 and 2 and the associated discussion into the results section

RESPONSE: We have decided to merge these tables and then present the implications on the decision-making process as an overview table in the results section. However, as the implications for informed decision-making is based on our own interpretation of the findings and not grounded in the data itself (the participants did not explictly reflect on how informed their decision was; we rather used references from the bioethics literature to make such interpretations), we found it more suitable to place these in the discussion section for clarity.

17. This section would benefit from a more comprehensive discussion of how findings deviate or align with existing work related to values and preferences related to hereditary cancer genetic testing. Specifically (and as mentioned in introduction comments) what is novel about this contribution to the evidence base and how can findings be used to inform practice/ policy/ future research?

RESPONSE: Thank you, we have restructured the discussion: we first align our findings with existing work and then focus on implications for informed decision-making, stressing more clearly that our inductive, context-specific approach is of practical relevance and the framing of informed decision-making is, to our knowledge, unique.

18. The following sentence requires clarification/ elaboration, “Because our results should inform genetic counseling practice, our results are still valid, even though from a conceptual perspective the view of those even refusing to go to genetic counseling would have been highly interesting.”

RESPONSE: Thank you, we have revised and clarified the meaning of our argument. It now reads: "Those counselees who completely refuse genetic testing might not even bother seeing a genetic counselor and are thus not included in this study, even though from a conceptual perspective these views might have been an interesting addition to our findings from a conceptual perspective. Yet, our findings are still relevant from a practical perspective, as they include views of those seeing a genetic counselor."

19. Pending elaboration on recruitment methods, the authors may need to include additional limitations related to their sampling and cessation of recruitment (e.g. why were 18 interviews conducted with 18 at risk individuals and only 2 with physicians)

RESPONSE: We have extended the limitations section according to potential shortcomings in theoretical saturation based on participant demographics.

---

## [Decision Letter · Decision Letter 1]

21 Jul 2021

PONE-D-21-09002R1

The use of heuristics in genetic testing decision-making: A qualitative interview study

PLOS ONE

Dear Dr. Zimmermann,

Thank you for submitting your manuscript to PLOS ONE. After careful consideration, we feel that it has merit but does not fully meet PLOS ONE’s publication criteria as it currently stands. Therefore, we invite you to submit a revised version of the manuscript that addresses the points raised during the review process.

Please submit your revised manuscript by 08-August-2021. Please include the following items when submitting your revised manuscript:

We look forward to receiving your revised manuscript.

Kind regards,

Prof. Ritesh G. Menezes, M.B.B.S., M.D., Diplomate N.B.

Academic Editor

PLOS ONE

Reviewers' comments:

Reviewer's Responses to Questions

**Comments to the Author**

1. If the authors have adequately addressed your comments raised in a previous round of review and you feel that this manuscript is now acceptable for publication, you may indicate that here to bypass the “Comments to the Author” section, enter your conflict of interest statement in the “Confidential to Editor” section, and submit your "Accept" recommendation.

Reviewer #2: All comments have been addressed

Reviewer #3: All comments have been addressed

Reviewer #4: (No Response)

2. Is the manuscript technically sound, and do the data support the conclusions?

Reviewer #2: Yes

Reviewer #3: Yes

Reviewer #4: Yes

3. Has the statistical analysis been performed appropriately and rigorously? 

Reviewer #2: N/A

Reviewer #3: Yes

Reviewer #4: N/A

4. Have the authors made all data underlying the findings in their manuscript fully available?

Reviewer #2: (No Response)

Reviewer #3: Yes

Reviewer #4: No

5. Is the manuscript presented in an intelligible fashion and written in standard English?

Reviewer #2: No

Reviewer #3: Yes

Reviewer #4: Yes

6. Review Comments to the Author

Reviewer #2: The comments were well answered. The paper is interesting and it is good to be able to publish and reflect on well explained qualitative research. The choice of "minor revisions" concerns a few points of English that are a bit awkward and a few typos. It's not a big thing and an additional proofreading by a native English speaker could improve the flow of lecture.

Reviewer #3: (No Response)

Reviewer #4: 1. The justification for two additional interviewees (experts) remains unclear. Although the authors have included the interview guide, expert quotes are not included in the results or highlighted in the discussion. The analytic approach to the expert interviews is not clearly reported.

2. My previous comment regarding a characterization in variation in preferences was not intended to suggest that the authors apply a quantitative approach to the analysis by reporting proportions. Rather, it would be helpful to provide a descriptive understanding of where opinions varied (or were consistent) across participants.

3. Exclusion of pregnant people remains unclear. If this was an issue for ethical approval, please estate that in the paper.

4. The authors appear to have uploaded multiple versions of the appending documents (interview guides)

7. PLOS authors have the option to publish the peer review history of their article (what does this mean?). If published, this will include your full peer review and any attached files.

Reviewer #2: No

Reviewer #3: No

Reviewer #4: No

---

## [Author Response · Author response to Decision Letter 1]

4 Oct 2021

We thank the two reviewers for their additional comments. We react to them in detail below.

Response to reviewers

Reviewer #2: The comments were well answered. The paper is interesting and it is good to be able to publish and reflect on well explained qualitative research. The choice of "minor revisions" concerns a few points of English that are a bit awkward and a few typos. It's not a big thing and an additional proofreading by a native English speaker could improve the flow of lecture.

Author response: We thank reviewer 2 for their positive response. An English native speaker with a degree in English language has revised the language of the paper.

Reviewer #3: (No Response)

Reviewer #4: 1. The justification for two additional interviewees (experts) remains unclear. Although the authors have included the interview guide, expert quotes are not included in the results or highlighted in the discussion. The analytic approach to the expert interviews is not clearly reported.

Author response: We thank reviewer 4 for their comments and clarifications. We highlighted the analytic approach in the methods section (page 10), it reads: “We conducted these interviews after an initial analysis of all counselee interviews and included them in the analysis to examine whether experts supported or contradicted or general understanding of interview data.” It also reads in the first section of findings: “Expert interviews were used to confirm and contrast counselees' views.” Along with this analytic approach, we repeatedly highlighted in the results where experts supported or contrasted our findings. For example, on page 13 it reads: “Conversely, one expert reflected on the difficulty of explaining these variants to counselees in the context of panel testing, stating that those lower risks or variants of unknown significance added complexity that was difficult to communicate. This expert feared that counselees would instead overestimate the risks of these variants.”

While we do indirectly cite experts in the results, we chose not to select direct quotes from experts. When selecting illustrative quotes, we attempted to illustrate our findings while maintaining a readable structure to the readers. We think that the quotes from counselees serve this purpose better. 

The reason why expert views are not highlighted in the discussion is because they mainly supported our findings. This was also why we stopped recruiting experts after two interviews. We did consider where we could include expert views in the discussion, but decided that it was beyond the scope to do this more prominently because we did not ask them directly about group A and B heuristics. 

2. My previous comment regarding a characterization in variation in preferences was not intended to suggest that the authors apply a quantitative approach to the analysis by reporting proportions. Rather, it would be helpful to provide a descriptive understanding of where opinions varied (or were consistent) across participants.

Author response: We thank reviewer 4 for the clarification. Indeed, our whole analytic approach aimed at examining variations and consistencies among counselees and experts. We report consistencies through the topics (heuristics) that were commonly reported, and then describe in detail the various views reported by participants. We clarified this in the first paragraph of the findings section (page 10).

3. Exclusion of pregnant people remains unclear. If this was an issue for ethical approval, please estate that in the paper.

Author response: It was indeed an issue of the ethics committee to not include vulnerable people, including pregnant women. We rephrased in the methods section, it now reads on page 8: “Counselees between the age of 18-70 that were not considered vulnerable due to mental state or pregnancy by the ethics committees that approved the study were eligible.» 

4. The authors appear to have uploaded multiple versions of the appending documents (interview guides)

Author response: This is because we used different interview guides for patient and experts – File S2 is the interview guide used for patients, File S3 is the interview guide used for expert interviews.

---

## [Decision Letter · Decision Letter 2]

15 Nov 2021

The use of heuristics in genetic testing decision-making: A qualitative interview study

PONE-D-21-09002R2

Dear Dr. Zimmermann,

We’re pleased to inform you that your manuscript has been judged scientifically suitable for publication and will be formally accepted for publication once it meets all outstanding technical requirements.

Kind regards,

Prof. Ritesh G. Menezes, M.B.B.S., M.D., Diplomate N.B.

Academic Editor

PLOS ONE

Reviewers' comments:

Reviewer's Responses to Questions

**Comments to the Author**

1. If the authors have adequately addressed your comments raised in a previous round of review and you feel that this manuscript is now acceptable for publication, you may indicate that here to bypass the “Comments to the Author” section, enter your conflict of interest statement in the “Confidential to Editor” section, and submit your "Accept" recommendation.

Reviewer #1: (No Response)

Reviewer #4: All comments have been addressed

2. Is the manuscript technically sound, and do the data support the conclusions?

Reviewer #1: Yes

Reviewer #4: Yes

3. Has the statistical analysis been performed appropriately and rigorously? 

Reviewer #1: N/A

Reviewer #4: N/A

4. Have the authors made all data underlying the findings in their manuscript fully available?

Reviewer #1: Yes

Reviewer #4: No

5. Is the manuscript presented in an intelligible fashion and written in standard English?

Reviewer #1: Yes

Reviewer #4: Yes

6. Review Comments to the Author

Reviewer #1: This was a well written and interesting manuscript reporting the findings of a qualitative analysis of the complexity of genetic testing decision making and the use of heuristics. The articulation of ways in which heuristics were useful and potentially not useful to counselees (Table 3) was appreciated and will likely be of use to clinicians. The authors appear to have addressed previous reviewer concerns and I raise no additional concerns.

Reviewer #4: Thank you for your careful responses to the suggested revisions. The authors have thoughtfully responded to each of my comments.

7. PLOS authors have the option to publish the peer review history of their article (what does this mean?). If published, this will include your full peer review and any attached files.

Reviewer #1: **Yes: **Fuchsia Howard

Reviewer #4: No

---

## [Editor Report · Acceptance letter]

18 Nov 2021

PONE-D-21-09002R2 

The use of heuristics in genetic testing decision-making: A qualitative interview study 

Dear Dr. Zimmermann:

I'm pleased to inform you that your manuscript has been deemed suitable for publication in PLOS ONE. Congratulations! Your manuscript is now with our production department. 

Kind regards, 

on behalf of

Prof. Dr. Ritesh G. Menezes 

Academic Editor

PLOS ONE